# Care to share? Experimental evidence on code sharing behavior in the social sciences

**Daniel Krähmer** [ID]*, **Laura Schächtele, Andreas Schneck**

Department of Sociology, University of Munich (LMU), Munich, Germany

* daniel.kraehmer@soziologie.uni-muenchen.de

## Abstract

Transparency and peer control are cornerstones of good scientific practice and entail the replication and reproduction of findings. The feasibility of replications, however, hinges on the premise that original researchers make their data and research code publicly available. This applies in particular to large-*N* observational studies, where analysis code is complex and may involve several ambiguous analytical decisions. To investigate which specific factors influence researchers' code sharing behavior upon request, we emailed code requests to 1,206 authors who published research articles based on data from the European Social Survey between 2015 and 2020. In this preregistered multifactorial field experiment, we randomly varied three aspects of our code request's wording in a 2x4x2 factorial design: the overall framing of our request (enhancement of social science research, response to replication crisis), the appeal why researchers should share their code (FAIR principles, academic altruism, prospect of citation, no information), and the perceived effort associated with code sharing (no code cleaning required, no information). Overall, 37.5% of successfully contacted authors supplied their analysis code. Of our experimental treatments, only framing affected researchers' code sharing behavior, though in the opposite direction we expected: Scientists who received the negative wording alluding to the replication crisis were *more* likely to share their research code. Taken together, our results highlight that the availability of research code will hardly be enhanced by small-scale individual interventions but instead requires large-scale institutional norms.

## Introduction

Transparency and openness are hallmarks of science. They are vital for the scientific enterprise to unfold its self-correcting capabilities and enable researchers to engage in "organized skepticism" [1] by evaluating and replicating scientific findings originally published by others. Such skepticism is warranted, given evidence of a replication crisis. Across disciplines, researchers have struggled to replicate key findings [2–6], meaning that central claims or discoveries could not be repeated independently. Against this backdrop, both conceptual and direct replications—the latter sometimes being called *reproductions* [7]—have been deemed crucial for determining whether findings are credible and deserve to enter the stock of scientific knowledge [8].

**Data Availability Statement:** All data and code underlying the results presented in this study are available from OSF (https://osf.io/t9p5v/?view_only=b3fde1fb9af14ce1b55c7a4b9741415e). To

safeguard participants' privacy, all publicly available data have been fully anonymized.

**Funding:** This research has been funded by the German Research Foundation (www.dfg.de/en) through the Priority Program META-REP (Project 464507200). The funders had no role in study design, data collection and analysis, decision to publish, or preparation of the manuscript.

**Competing interests:** The authors have declared that no competing interests exist.

In practice, reproductions hinge on the availability of an original study's code and data. Performing a replication solely based on the body of an article is tedious at best and impossible at worst. This goes without assuming ill intent or incompetence on the authors' side: Space in research articles is limited, and even the most diligent researcher will struggle to document all relevant details of their analysis within an article.

To boost transparency, publishing research data has become increasingly common, not least due to requirements imposed by funders (e.g. by the European Research Council). While publishing research data is clearly a step towards more open science, it is only half the battle. Given the complexity of analyses, a dataset may yield very different conclusions depending on how it is processed and analyzed. This has been demonstrated by a number of many-analysts studies [9–11] which sparked lively debate across the social sciences [12, 13]. Researchers' degrees of freedom imply that, in extreme cases, open data may provide little to no extra transparency if replicators cannot reconstruct data preparation and analysis procedures. This applies in particular to large-$N$ observational studies, where analytical flexibility and errors typically unfold downstream of data collection (i.e. sample restrictions, outlier management, coding of missing values). To facilitate effective peer control, open data thus needs to be accompanied by open code (i.e. Stata do-files, SPSS syntax, R scripts).

Open code alleviates two problems that currently compromise the credibility of research: Errors in data preparation and misspecifications of statistical models [8]. While model misspecification should, in principle, be discernible from the body of an article, errors in data preparation are downright impossible to spot without access to authors' code. In fact, even wrong model specifications may be hard to detect, given that authors rarely justify their statistical models sufficiently for replicators to exert effective scrutiny [14]. Open code advances research integrity by exposing errors and lifting the lid on opaque model descriptions.

Fostering research transparency is not the only benefit of open code. Code quality itself may benefit from being subject to public scrutiny, if only through the short-term motivation of making one's syntax readable and understandable to others [15]. Code sharing may also increase efficiency, as not all scientific endeavors require reinventing the wheel [16, 17]. Ultimately, re-using and adapting peers' code saves time and valuable resources, thereby facilitating peer learning and advancing the progress of science.

## State of the art and research question

Despite the emergence and rapid dissemination of online repositories such as OSF and Github [18], open code remains the exception rather than the rule for most published research articles. As a consequence, replicators routinely find themselves forced to contact original authors, relying on the latter's spontaneous willingness to cooperate. Critically, researchers have repeatedly proven reluctant to comply with such requests. In an early study, Wolins [19] found that out of 37 psychologists, only 24% responded positively to a student's query for raw data and analysis material. Recent studies report similar sharing rates, usually ranging from around 20% [4, 20] to 45% [21, 22]. To tackle the unavailability of research material, some journals (i.e. by the American Economic Association) have recently started requiring authors to provide all data and code files upon submission. These replication packages are then checked by a designated data editor to ensure reproducibility [23].

Fig 1 summarizes the existing literature on data and code sharing (see also Table A in S1 File). Three details are noteworthy. First, even the most optimistic studies [20, 24] do not yield sharing rates above 60%, underpinning the assertion that current research falls far short of full transparency. Second, studies outside the common 20–45% sharing range rely on small samples (see marker size), yielding less precise estimates. Third, researchers' reluctance to share

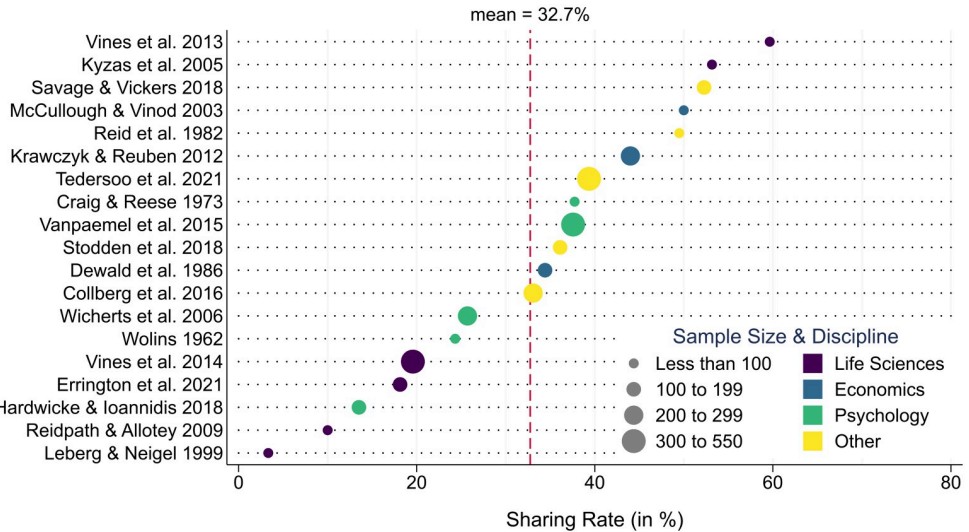

**Fig 1. Researchers' willingness to share data/code in the literature.** Note: Dot-plot of data/code sharing rates in the literature [4, 19–21, 24–38]. Sample sizes of the respective sharing studies are denoted by dot size. The discipline under study is represented by different dot colors.

materials transcends disciplinary boundaries. Providing data and code upon request has proven equally unpopular in the life sciences, economics, psychology, and various other disciplines.

Although there is convincing evidence that scientists' willingness to share research material is generally low, little is known *why* researchers choose to withhold or release their code. Correlational studies have established associations between authors' willingness to share research material and a study's age [20] as well as its strength of evidence [22]. While both findings offer tempting explanations, i.e. code and data get lost over time and authors actively obstruct verification attempts of shaky results, neither allows for straightforward causal interpretation: The seeming age effect might simply mirror the tightening of journals' data policies over time. Similarly, the link between an article's statistical properties and material availability may be confounded by researchers' self-selection into particular fields and topics.

In a rare attempt to provide causal evidence on knowledge sharing in academia, Krawczyk and Reuben [21] conducted a field experiment asking 200 authors to share supplementary material for published articles. Requests were sent from either Krawczyk's or Reuben's institutional email account and included the requestor's name and affiliation (University of Warsaw or Columbia University, respectively). Additionally, half of all emails identified the sender as an assistant professor. Exploiting this experimental variation, Krawczyk and Reuben found that neither the requestor's affiliation nor academic position strongly impacts response and compliance among contacted authors. While these results are soothing from an egalitarian perspective, they come with two major limitations. Regardless of their experimental condition, participants in the experiment could easily check the requestor's academic position online, potentially weakening the treatment effect. Furthermore, email signatures always included both the requestor's name and institutional affiliation, possibly conflating two analytically distinct treatments.

To this day, the factors influencing scientists' code sharing behavior remain unknown. What facets of a code request may favor or impede original authors' willingness to share code? And, by extension, (how) can replicators leverage this knowledge to increase the availability of

research code through micro-interventions? Our analysis presents a theory-guided attempt to answer these questions.

## Theoretical background

Invoking the fundamentals of rational choice and game theory, we conceptualize code sharing in academia as a public good game [39–41]. Collectively, open code is desirable as it fosters the expansion and consolidation of common knowledge by increasing research efficiency and facilitating peer-control. Individually, however, researchers are subjected to academia's "publish-or-perish" precept [42, 43] which yet gives little to no credit for subsidiary research output, such as analysis code, in the pursuit of tenure and reputation [44, 45]. Rational researchers who aim to maximize personal benefits while minimizing costs (i.e., investment of time and resources, risk of repercussions from peer-control) will refrain from contributing to the public good, running headfirst into a social dilemma [39, 46, 47]. As a result, all researchers lack the collective good.

We build upon this understanding of code sharing as a social dilemma and examine the effect of nudges on increasing cooperation in the academic public good game [48–50]. Nudges are deliberate yet subtle modifications to an individual's "choice architecture", i.e., the environment in which they make a decision [51]. They are low-cost, easy-to-miss micro-interventions that do not restrict the nudgee's freedom of choice but may steer them towards specific decisions. While most prominently discussed as a method to nudge individuals towards acting in their own self-interest (e.g., by making healthier food choices), nudges have also been discussed as a means to foster prosocial behavior in social dilemmas, such as vaccine uptake [52] and pro-environmental behavior [53, 54].

We focus on three types of nudges that may reasonably affect researchers' behavior: the framing of our code request, salience nudges towards its implied appeal or benefit, and the friction associated with compliance. The concept of framing assumes that even seemingly trivial changes in the formulation of choice problems can significantly alter people's preferences and behavior [55]. For public good games, evidence suggests that participants are generally more likely to cooperate if a choice situation is framed positively [56–58]. We expect this also applies to code sharing and formulate a framing hypothesis.

$H_1$: Authors are more likely to share code if the request is framed positively (i.e., to advance science) compared to a negative framing (i.e., in response to the replication crisis).

Salience nudges are commonly employed to steer attention to factually correct aspects, potentially relevant for the individual decision, and yet easily overlooked [59]. Against the backdrop of rational choice theory, we expect that the salience of specific benefits may increase the likelihood of code sharing among researchers. As code sharing has rarely been investigated as a distinct phenomenon, we pertain to survey research on attitudes towards data sharing to identify three potential appeals: compliance with common norms [43, 60], altruistic preferences [43, 61] and actual career benefits via rightful acknowledgment [43, 62–64]. We propose that compared to the untreated control group, researchers should be more likely to share code if our request highlights. . .

$H_{2.1}$: . . .that code sharing is in compliance with normative, institutional-level open science norms, such as the FAIR Guiding Principles [65].

$H_{2.2}$: . . .the importance of authors' participation for the scientific community (i.e., academic altruism).

$H_{2.3}$: . . .positive effects of code sharing on authors' own careers, i.e., by future citation.

Considering the opposite end of researchers' cost-benefit analysis, we argue that short-term costs associated with code sharing (i.e., locating, commenting, and preparing code) are crucial for researchers' decisions. Indeed, previous research on data sharing has repeatedly pointed towards the investment of time and effort as major disincentives [66, 67]. Assuming rational researchers strive to minimize costs, we posit an effort hypothesis.

$H_3$: Researchers are more likely to share code if we point out that code cleaning is not required (compared to the untreated control group).

## The present study

The article at hand presents the results of a large-scale, fully randomized field experiment. We aim to investigate which facets of a code request influence researchers' code sharing behavior and thereby extend previous research in three fundamental ways. First, whereas prior studies mainly provided correlational evidence, our experimental design enables us to identify causal predictors of code sharing among peers. Second, we are among the first to acknowledge *code* sharing as a distinct phenomenon, separate from *data* sharing. This differentiation is crucial as code sharing involves a very different set of practical, ethical, and legal considerations (i.e. file size, privacy issues, copyright). Third, our study descriptively sheds light on the availability of research code in the social sciences. Despite a considerable body of research from psychology and economics, little is known about code sharing in neighboring disciplines such as sociology and political science.

All hypotheses and the research design of this study have been preregistered (https://osf.io/bqjcz). Deviations from the preregistration occurred in four very minor instances (e.g. adapting the language of correspondence based on authors' reply). All deviations are listed, described and justified in Table B in S1 File.

## Materials and methods

We sent code requests to 1,206 authors who published research articles based on data from the European Social Survey (ESS) between 2015 and 2020. The ESS is a high-quality, biannual cross-national survey commonly used in the social sciences. Restricting our sample to ESS analyses offers three advantages. First, ESS data is publicly available, meaning researchers may not invoke privacy concerns as a justification for not sharing their analysis code. Second, ESS users stem from various social science domains, strengthening the external validity of our results. Third, the ESS administrative team curates a list of research articles that rely on ESS data, offering a comprehensive and clear-cut target population for our study.

### Sampling procedure

The complete ESS bibliographic database obtained from the ESS administrative team contained 5,429 entries. We restricted our sample to journal articles published between 2015 and 2020 to ensure that original authors could still be expected to have the code for their analyses available. We excluded working papers and theses which are predominantly written by students and early career researchers, as well as book sections. Focusing on journal articles allows us to distinguish research quality and code sharing, as journal articles have passed at least some quality control in the peer-review process. Furthermore, we excluded duplicate entries from the database and articles for which full texts were not available. We assumed the corresponding author to be the most natural point of reference for our code request. If this information was not included, we used available contact details for any author in order of appearance. If an author from our contact database had published multiple eligible articles, we randomly

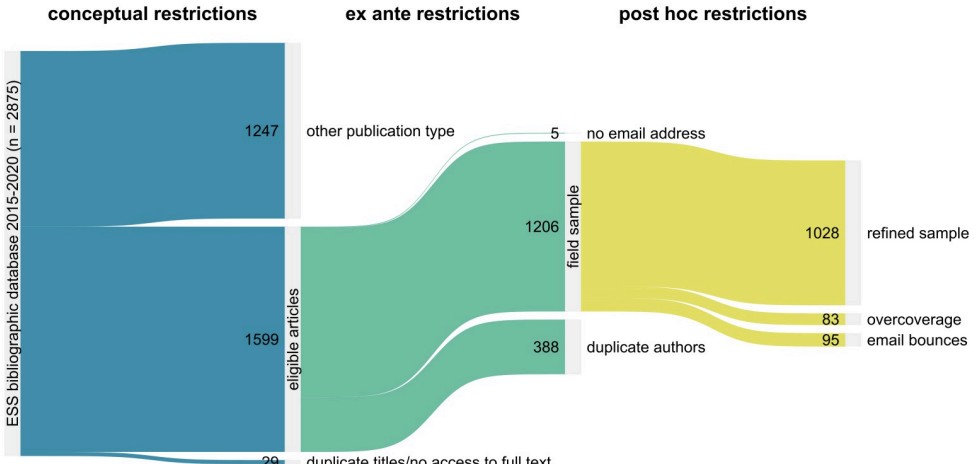

**Fig 2. Sampling procedure starting from the ESS bibliographic database.** Note: Sankey-plot of the sampling procedure. Conceptual as well as ex ante restrictions are described in the preregistration ($N = 1,206$). Post hoc restrictions were deemed necessary during the field phase to reduce overcoverage and correct for sample-neutral failures ($N = 1,028$). This graph has been created using a modified version of the Stata ado `sankey` [68].

chose one publication to minimize individual burden and avoid reactivity, as researchers may have uncovered that they took part in an experiment. Implementing all sample restrictions left us with 1,206 cases. Fig 2 illustrates the sample selection process in detail.

Given the increasing popularity of online repositories, we expected some authors to have made their code publicly available prior to our request. To check this, we automatically screened all articles in our final sample for hyperlinks to scientific repositories. We found references to online material in 39 cases (3.2%). Upon manual inspection, some of these readily available replication packages seemed fragmentary or insufficiently documented, which led us to contact the authors nonetheless.

## Field phase

On 6 and 7 July 2022, we reached out to all 1,206 authors in our final sample via email. Requests were sent from an institutional email address affiliated with the department of Sociology at LMU Munich and were signed by D.K. 221 emails (18.3%) bounced due to invalid email addresses. In such cases, we manually searched for alternative email addresses online and contacted the authors the next day. If no such address could be obtained or if the second email, too, resulted in a bounce, we ultimately deemed that request unsuccessful ($n = 95$). To increase response, we sent up to three reminders at two-week intervals, the first and second of which reiterated our experimental treatments. Whenever any of our emails triggered an out-of-office notification, we stalled all further dispatches until the author's return. If researchers referred us to one of their coauthors, all subsequent correspondence was redirected accordingly. The field phase ended on 17 January 2023. Approval to conduct this study has been granted by the Institutional Review Board of the Faculty of Social Sciences at LMU Munich (GZ 22–03). Due to the study's field experimental setting, informed consent could not be obtained prior to individual study participation [69]. Following the recommendations by the ethics board, all authors who had responded to any of our emails were debriefed on 17 January 2023 regarding the experimental nature of our request unless they had explicitly objected to receiving any further emails. Fig A in S1 File provides a breakdown of the daily inflow and outflow of emails between 6 July 2022 and 17 January 2023.

## Experimental design

To investigate the determinants of researchers' code sharing behavior, we experimentally varied three aspects of our request's wording.

- **Framing of the project (2 levels)**: The positively worded version of our email stated that researchers' cooperation would enhance the quality, relevance, and success of social science research. The negative version framed our project in light of the replication crisis and its ramifications for the credibility of social science research.

- **Appeal of code sharing (4 levels)**: The baseline version of our request did not include any specific appeal on why researchers should share their code. Other versions stated that by sharing code, researchers would either i) honor the FAIR Guiding Principles [65], ii) commit an act of academic altruism by helping other researchers, or iii) increase their chances of being cited as part of our replication attempt.

- **Perceived effort (2 levels)**: The reduced effort version of our request emphasized that authors were not expected to clean their code before sharing. No such statement was included in the baseline request.

This 2x4x2 factorial design resulted in 16 randomly assigned treatment conditions ($n \approx 75$ each). Wherever possible, we kept the wording of treatments at a comparable length to avoid confounding (e.g. by lengthy requests lowering researchers' willingness to share). We did not vary the order of the three treatment dimensions to ensure readability and coherence. Besides the varied treatment dimensions, the request provided general background information on our research project and stated our interest in replicating published findings. Fig B in S1 File provides an exemplary email; all email templates are available in the preregistration form (https://osf.io/bqjcz).

The central outcome variable was dummy coded based on the correspondence with the authors (0: Code not shared; 1: Code shared via email/link to online repository). All analyses are based on the authors' sharing behavior upon completion of the field experiment in January 2023. Researchers who promised to share their code but failed until our deadline were coded as non-compliant ($n = 28$). This seems reasonable, given that potential replicators should not have to wait more than 6 months to receive code from the original authors. As the research design required repeated correspondence, it was inevitable that we were able to identify individual participants during data collection and analysis. To safeguard participants' privacy, all publicly available data have been fully anonymized.

We employed two-sample $z$-tests of proportions to test our experimental treatments. Due to the experimental design and the random allocation of treatments, this approach yields an unbiased estimation of our treatment effects. Assuming a moderately large effect, i.e. our treatment leading to an increase in code sharing rates by 15 percentage points ($d = 0.3$), the statistical power for both two-level and four-level treatments is more than adequate (96.25% and 99.93%, respectively; see Text A in S1 File for a detailed discussion on statistical power). As all our preregistered hypotheses are directional, we used one-tailed tests to evaluate the z-scores. All calculations have been performed using Stata version 17.0.

## Results

### Descriptive

Overall, 658 (59.2%) out of 1,111 successfully delivered emails (final sample net of bounced emails) triggered a response to our request. Some authors claimed their article would fall outside our sampling frame as their analyses did not use ESS data substantially. If this proved to

be true upon manual inspection, cases were coded as ineligible due to overcoverage and excluded from our analyses ($n = 66$). If an article did report results based on ESS data despite an author's contrary claim, we followed-up and affirmed our interest in the research code. In certain instances, we detected author duplicates after the initial emails had already been sent (e.g. if an email was forwarded to an author already included in our sample). To minimize the workload for each author and reduce reactivity, we randomly selected one article from these duplicates and requested replication material for this selected article in subsequent emails ($n = 16$). In one case, we inadvertently contacted a namesake. Applying all post hoc restrictions left us with a refined sample of 1,028 cases. Among these eligible cases, we obtained a response from 56.7% of the authors ($n = 583$).

To our surprise, a sizable share of researchers indicated to be unfamiliar with the concept of research code. This finding may, on the one hand, be attributable to terminological differences across disciplines and software packages ("code", "syntax", "script", etc.). It might, on the other hand, reflect the persistence of point-and-click solutions, which do not require writing reproducible code in the first place. If authors indicated confusion about what we meant by research code ($n = 22$), we sent a follow-up email providing examples and clarification.

Upon completion of our field experiment, we obtained research code for 385 articles. This amounts to an aggregate sharing rate of 37.5% (of 1,028 eligible cases), which is largely in line with findings from previous research (see Fig 1). While 43.3% of researchers ($n = 445$) ignored our repeated attempts, 12.9% of authors ($n = 133$) stated upfront that they were unable or unwilling to share code. As Fig 3 illustrates, we received most code in response to our initial request ($n = 177$). Over the course of the three subsequent reminders, the number of shared code packages declined steadily ($n_{R1} = 77$, $n_{R2} = 70$, $n_{R3} = 61$). Among those who shared, the median timespan until we received the code files was 16 days (mean = 27; $SD = 29.9$).

Even on the surface, the code packages we received exhibit widely different features. An obvious, though shallow, indicator for this is the number of files per replication package. While most packages consisted of up to five files, 10.1% included 50 or more files (see Fig 4, Panel A). In two extreme cases, packages contained more than 10,000 files, suggesting that some authors interpreted our request more broadly and shared not only research code but also automatically generated output files (e.g. from simulation studies). Despite such nuances in understanding, the observed variance in file counts may reflect genuine differences in coding practices (e.g. bundling code vs. breaking code into task-specific files).

To assess software preferences, we automatically extracted the file extensions (e.g. ".dta", ".txt") of all shared files and created binary indicator variables for those unequivocally associated with certain statistical software (e.g. ".do" for Stata, ".r", ".sps", etc.). Analyzing these indicators reveals that the majority (58.4%) of sharing authors relies on Stata for statistical analyses, making it by far the most popular software in our sample (see Fig 4, Panel B). Furthermore, 46

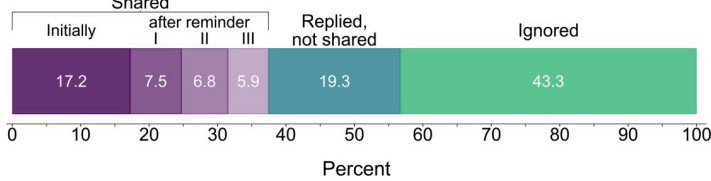

**Fig 3. (Non-)Response to our code request.** Note: Stacked-bar graph of sharing outcomes. Purplish colors denote the proportion of shared code in the refined sample ($N = 1,028$). Greenish colors denote the proportion of requests where no data was shared.

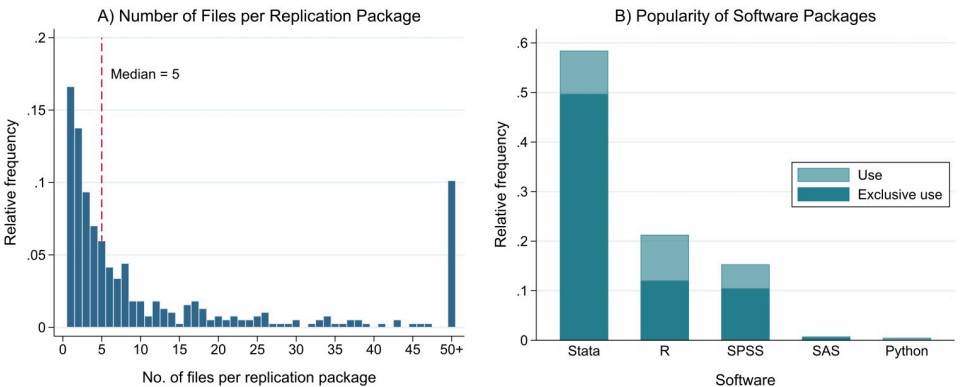

**Fig 4. Content of replication packages shared upon request.** Note: Descriptives for shared replication code. Panel A shows the number of files per replication package. Panel B shows the proportion of used software packages by the articles.

studies (11.9%) appear to use more than just one software, which is surprising given that obtaining proficiency in multiple coding languages requires a greater time investment from researchers. Although we can only speculate about the reasons for this pattern, two plausible explanations come to mind. Either researchers deliberately combine different software to exploit the strong points of every single program, or the mix of statistical software stems from collaborations of differently trained researchers.

Authors' replies to our code request can also be leveraged to gain insight into obstacles commonly associated with sharing one's analysis code. To obtain information on such hurdles, our third reminder explicitly asked non-compliant researchers to clarify why they could not share their research code with us. While some justifications were generic ("this research was done a long time ago"), others pointed towards concrete and systemic problems: 23 researchers admitted not having created any permanent code in the first place; 20 authors had already lost access to their research code, mostly due to changes in affiliation; 16 researchers reported hardware problems, ranging from corrupt hard-drives to large-scale hacker attacks; 15 authors confessed not being able to locate their code; and 12 individuals reported a lack of time. Although this snapshot is selective, it highlights that many obstacles to code sharing could be overcome by embracing a more institutional, centralized approach to archiving research code. Importantly, only a tiny minority of researchers ($n$ = 4) indicated to hold an explicit conviction of not sharing research code. In most cases, authors' responses were remarkably favorable, even though, in some cases, helpless.

## Main results

Fig 5 depicts code sharing rates across the main experimental treatments. Contrary to our hypothesis, we find a lower proportion of shared code among researchers who received the positively framed request. Invoking the replication crisis instead of emphasizing the quality and success of social science research appears to *increase* authors' inclination to share research code ($z$ = 2.287, $p$ = 0.989). Referencing the FAIR data principles ($z$ = -0.269, $p$ = 0.394), calling on researchers' altruism ($z$ = 1.180, $p$ = 0.881), and mentioning the prospect of future citation ($z$ = -0.077, $p$ = 0.470) does not impact code sharing behavior compared to the neutral baseline condition. Similarly, no differences arise when authors are exempted from code cleaning compared to the neutral control condition without such a statement ($z$ = -0.064, $p$ = 0.474). Full results for all confirmatory tests are reported in Table C in S1 File.

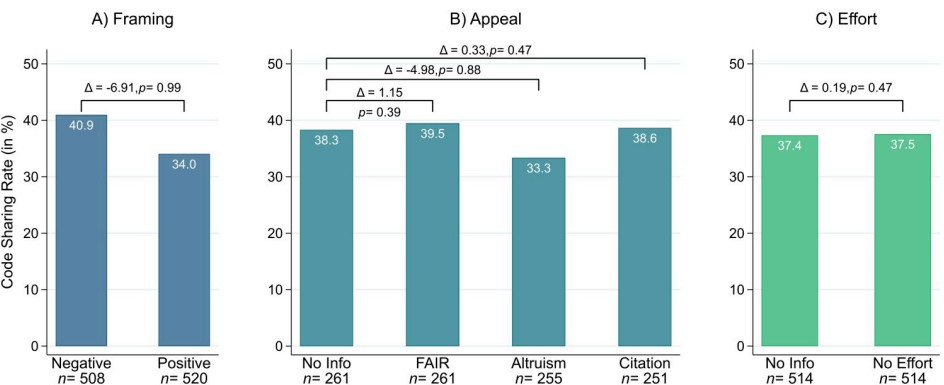

**Fig 5. Code sharing rate by main experimental treatments.** Note: Bar chart for code sharing rates across main experimental treatment effects. Brackets indicate differences and corresponding *p*-values across levels.

The seemingly positive impact of framing science as *in crisis* on code sharing behavior becomes visible also in Fig 6. As indicated by the first and second row of the figure's bottom panel, there is a clear separation between high returns (negatively framed) and low returns (positively framed). No such pattern emerges for the other experimental treatments. Comparing code sharing rates across all 16 unique treatment combinations reveals considerable variation in authors' behavior. As Fig 6 shows, code sharing rates ranged from about 20% to 50% across unique treatment conditions (for a depiction of *response* rates across treatment conditions, see Fig C in S1 File). For instance, positively framed requests that appealed to authors' altruism and mentioned that code cleaning was not required stimulated only 21.9% of authors to share their code with us (compared to 52.4% positive replies among those that received a negatively framed request which highlighted the FAIR norms and did not comment on code cleaning requirements).

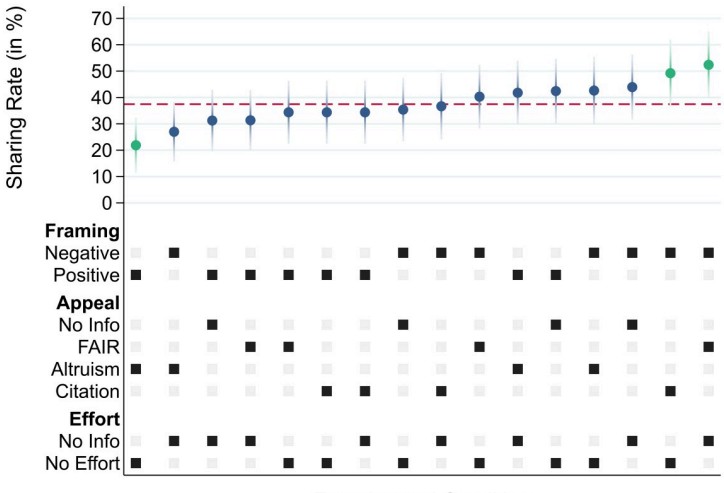

**Fig 6. Code sharing rates across all 16 unique experimental conditions.** Note: Code sharing rates for all 16 distinct treatment conditions with 95% confidence intervals. The treatment conditions are sorted in ascending order. Black/gray squares represent active/passive treatment conditions, respectively. Results from a linear probability model with all two-way interactions are reported in Table D in S1 File. This graph has been created using the Stata ado `mfcurve` [70].

## Discussion

This study provides a large-scale assessment of researchers' code sharing behavior upon request. With an overall sharing rate of 37.5%, our descriptive results align with previous research and demonstrate that code sharing is not common even among researchers who use publicly available observational data. Contrary to our preregistered hypotheses, framing our request positively did not increase code sharing among researchers. Conversely, we find higher code returns among researchers who received the negatively framed request alluding to the replication crisis' ramifications. Originating from psychology, the replication crisis has sparked discussion on good research practice across disciplines, emphasizing the need for open science [71, 72]. As such, it might have become a buzzword triggering researchers to comply with our request. Neither nudges towards potential appeals nor a reduction of the perceived effort had an effect on authors' likelihood of sharing their code. While these factors may influence researchers' *attitudes* on sharing their material [43, 66, 67], actual *behavior* seems to hinge on the prevailing incentive structures in academia. Considering the persistent lack of acknowledgment for the publication of subsidiary research output and the resulting unfavorable cost-benefit-ratio, simple nudges are clearly not sufficient to influence researchers' code sharing behavior. Our estimates might represent a somewhat lower bound of the true nudging effect, though, as some participants could have been impervious to nudging *per se*. This applies, for instance, to researchers engaging in questionable research practices who would have ignored our request regardless of its wording. We expect the number of duplicitous researchers in our sample to be low but have no way of testing this assumption. Importantly, in some cases the provision of code turned out to be impeded by the simple fact that such code was never written in the first place. This may be due either to the use of graphical user interfaces (GUI) in statistical analysis tools or to a lack of knowledge about data and code management. Unlike ignorance towards best practices of data and code management, the use of GUI solutions is not inherently problematic as long as sufficient additional information is provided to maintain analytic transparency.

Building on prior research findings, the main advantage of our study lies in its experimental design, which enables us to identify the causal effects of specific nudges on researchers' actual code sharing behavior. Choosing a large-*N*, multidisciplinary sample of journal articles using ESS data bolstered the external validity of our experiment. As ESS data is publicly available, it also enabled us to investigate code sharing as a distinct phenomenon. Nonetheless, our approach has its limitations. As our data fully relies on our email correspondence with authors, we were only privy to information they chose to share with us. Thus, some of our requests might not have been actively ignored but rather have fallen victim to spam-filters. Reassuringly, this should affect all treatment conditions equally, therefore not inducing bias to our experimental results. It may, however, lead to a slight underestimation of our descriptive sharing rate. Given the high response rate to our request (56.7% among eligible researchers), we remain confident, though, that such distortion is small. As the ESS bibliographic database does not provide information exceeding the individual bibliographic entry, we furthermore cannot assess the sample distribution of properties such as authors' career level or regional affiliation, both of which have been linked to attitudes and practices regarding open science [64, 73–75]. Therefore, we cannot rule out treatment heterogeneity in our experimental results (e.g., more experienced researchers being more strongly affected by the effort treatment). Authors who received multiple requests from our project (e.g. due to forwarding from other researchers) pose another potential threat to the validity of our conclusions, as these individuals might have been affected by reactivity. Excluding such cases from our analysis does, however, leave results unchanged (see Table E in S1 File).

Our findings provide fertile ground for further research on code sharing in academia. Most likely, authors' readiness to share code depends on several contextual factors such as career status, affiliation, as well as journal properties such as impact factor, and policy control. While such context variables are interesting to consider, none of them is exogenous and, as such, they fall outside the scope of our experimental research design. Overall, the present study points towards the limited effectiveness of individual micro-interventions and highlights the dire need for institutional solutions regarding code availability. To ensure transparency and bolster scientific credibility, requiring open code as part of any research submission should become the institutional standard [32, 72, 76, 77].

## Supporting information

**S1 File.**
(PDF)

## Acknowledgments

We thank Katrin Auspurg, who has been intimately involved in planning and conducting the experiment. Thanks to two anonymous reviewers for their helpful comments and to Richard Vielberg for superb research assistance. Lastly, we are indebted to all researchers who participated in our experiment and went to great lengths to share their research code with us.

## Author Contributions

**Conceptualization:** Daniel Krähmer, Laura Schächtele, Andreas Schneck.

**Data curation:** Laura Schächtele.

**Formal analysis:** Daniel Krähmer, Laura Schächtele, Andreas Schneck.

**Funding acquisition:** Andreas Schneck.

**Investigation:** Daniel Krähmer, Laura Schächtele, Andreas Schneck.

**Methodology:** Daniel Krähmer, Laura Schächtele, Andreas Schneck.

**Project administration:** Andreas Schneck.

**Software:** Daniel Krähmer.

**Visualization:** Daniel Krähmer.

**Writing – original draft:** Daniel Krähmer, Laura Schächtele.

**Writing – review & editing:** Daniel Krähmer, Laura Schächtele, Andreas Schneck.

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
