## [Decision Letter · Decision Letter 0]

17 May 2023

PONE-D-23-09667Care to share? Experimental evidence on code sharing behavior in the social sciencesPLOS ONE

Dear Dr. Krähmer,

Thank you for submitting your manuscript to PLOS ONE. After careful consideration, we feel that it has merit but does not fully meet PLOS ONE’s publication criteria as it currently stands. Therefore, we invite you to submit a revised version of the manuscript that addresses the points raised during the review process. Overall, the reviewers are very positive on your work but they ask for some precisions or alternative presentations of the results to clarify some points.

We look forward to receiving your revised manuscript.

Kind regards,

Simon Porcher

Academic Editor

PLOS ONE

Journal Requirements:

"This research has been generously funded by the German Research Foundation through the Priority Program META-REP (Project 464507200). We thank Katrin Auspurg, who has been intimately involved in planning and conducting the experiment. Thanks to Richard Vielberg for superb research assistance. Lastly, we are indebted to all researchers who participated in our experiment and went to great lengths to share their

research code with us"

"This research has been funded by the German Research Foundation (www.dfg.de/en) through the Priority Program META-REP (Project 464507200). The funders had no role in study design, data collection and analysis, decision to publish, or preparation of the manuscript."

Reviewers' comments:

Reviewer's Responses to Questions

**Comments to the Author**

1. Is the manuscript technically sound, and do the data support the conclusions?

Reviewer #1: Yes

Reviewer #2: Yes

2. Has the statistical analysis been performed appropriately and rigorously? 

Reviewer #1: Yes

Reviewer #2: Yes

3. Have the authors made all data underlying the findings in their manuscript fully available?

Reviewer #1: Yes

Reviewer #2: Yes

4. Is the manuscript presented in an intelligible fashion and written in standard English?

Reviewer #1: Yes

Reviewer #2: Yes

5. Review Comments to the Author

Reviewer #1: This is a review for "Care to share? Experimental evidence on code sharing behavior in the social sciences." The submission contains a 2x4x2 audit-style nudge experiment designed to elicit the reasons why authors of research using observational data do or do not share their code.

My recommendation for this experiment is a positive revise and resubmit. I consider the questions considered (the sharing of code, distinct from data sharing, replicability) and the methodology (audit study, causal interpretation, multiple treatment arms, sufficient power) to be of acceptable quality. I find solely targeting research articles using a single dataset is an important innovation, and a sample size of 1206 responses is large. I am further encouraged by the fact this analysis was pre-registered.

My comments below are split into major and minor.

Major Comments

1. Please include a discussion of the duplicitous researcher. If I am a duplicitous researcher, and I receive an email (as described) stating the following: "our project aims to assess the reproducibility of randomly selected articles from the European Social Survey’s bibliographic database. Would you mind sharing your code with us to make sure your article can be included in our analysis?" I will simply ignore it. What implications does this have for the paper's results? To my reading, this will mean that those who reply are more likely to be honest researchers. The estimated nudge effects are then lower bounds, should the proportion of duplicitous researchers shrink in the future (who would not be nudge-able ever).

2. While Table D of the supplementary information is almost comprehensive, I would like to see the following: Line 234 mentions a strong assumption. " Researchers who promised to share their code but failed until our deadline were coded as unwilling to share." Are the results robust to their exclusion? Are results robust to their coding as 1? This proportion of researchers seemed non-trivial.

3. Please answer: By treatment, is response rate different? I understand that the final dummy variable is 0 when not and 1 when the code was shared, but response rate could tell the reader something more. The data is already well anonymized so I do not see this being an issue of privacy for this manuscripts authors.

4. What is the proportion of authors that needed clarification about what research code is? Line 268. Is this extra coaching applied evenly between treatments?

5. There is an assumption being made here about no interactivity between nudges. I would like to see a simple regression examining this, using interaction effects. A simple linear probability model would suffice in my opinion.

6. The post-hoc achieved power for the framing treatment is 62.7%. However, this includes non-response. From G*Power we have...

z tests - Proportions: Difference between two independent proportions

Analysis: Post hoc: Compute achieved power

Input: Tail(s) = Two

Proportion p2 = 0.409

Proportion p1 = 0.340

α err prob = 0.05

Sample size group 1 = 508

Sample size group 2 = 520

Output: Critical z = 1.9599640

Power (1-β err prob) = 0.6277340

For the framing statistical test. I would like to see calculations of post-hoc achieved power for the remainder of the hypotheses.

Minor Comments

1. I downloaded and was able to successfully replicate the figures and tables using the same code and data as provided by the authors. The data has a last modified date of 2023-03-30. I then followed up on the pre-registration on "" ext-link-type="uri" xlink:type="simple">https://osf.io/bqjcz". The registration was created on 2022-07-06. The only unfortunate thing I can note (and this fact is by no means unique to this project) that it is not possible for the reviewer to verify if the registration was done prior to sample collection with the replication package as is. To this end, I must attach less meaning to the pre-registration than I would be able to otherwise. This can be remedied easily. Line 188 of the manuscript does mention that data collection began on 2022-07-06. I would appreciate some/any method of verification of the emails / replies being sent / coming in after the pre-registration was released.

2. Quotation marks, particularly opening quotations are often "backwards".

3. Line 13. I believe on line 13 the authors mean "on the authors' side". Typo in line 303 "a long time ago"

4. Line 20. The following should be mention/cited here, as it is a well-known and publicized example of this.

Silberzahn, R., Uhlmann, E. L. (2015). Crowdsourced research: Many hands make tight work. Nature, 526(7572), 189-191.

Silberzahn, R., Uhlmann, E. L., Martin, D. P., Anselmi, P., Aust, F., Awtrey, E., ... Nosek, B. A. (2018). Many analysts, one data set: Making transparent how variations in analytic choices affect results. Advances in Methods and Practices in Psychological Science, 1(3), 337-356.

Reviewer #2: This is an interesting and very innovative paper on an important topic. In a large randomized field experiment, the authors investigate how they can get researchers to share their analysis code upon requests. Very neatly, the authors only contact researchers who have worked with data from the European Social Survey, which means that data sharing is not an issue. That’s really great! In this pre-registered experiment, the authors randomly vary various aspects of the code requests, and largely find null effects of their treatments with the exception of a positive effect of negative wording alluding to the replication crisis and this was against the prior of the authors.

I am very positive to this paper. It was great to see the actual emails sent out as this makes it clear to me that the message must have come across to those that read the email. Most of my comments are minor and are written below in the order of appearance in the paper:

p.2: You write “In practice, replications hinge on the availability of an original study’s code and data.” I would argue that this is about reproducibility, and that replication involves new data. I think this is how most psychologists would define replication (including the work of Brian Nosek and others). Some discussion of reproducibility vs replicability might make sense to include.

p.2: When writing about publishing research data being a step towards more open science, I guess this also depends on how “true” we think the data is in terms of whether we only get to see the p-hacked data or not (i.e. outcome variables that were not significant are not included etc).

p.3, first paragraph in “state of the art”: Here it could be interesting to note that journals increasingly have Data Editors (at least in economics and political science) who go through the data and code to make sure that the code runs and leads to the same results as in the paper. For example, the journals of the American Economic Association only conditionally accept papers before the data editors have approved the data and code, which timing-wise makes sense.

Theoretical background: Do you consider all framing nudges? I personally do not really see the need to call the different versions in your experiment nudges, but this is of course up to you. When citing references 49-51 it looks in the main text like those studies were on code sharing PGGs which is not the case since they are on standard ones – perhaps that could be clarified?

In relation to H2.3: Perhaps a bit related to this is the paper by Christensen et al. who find that data sharing is positively related to citations (Christensen G, Dafoe A, Miguel E, Moore DA, Rose AK (2019) A study of the impact of data sharing on article citations using journal policies as a natural experiment. PLoS ONE, 14(12): e0225883. https://doi.org/10.1371/journal.pone.0225883).

p.7: I very much appreciate the transparency in how deviations from the pre-analysis plan are listed. From my understanding of reading them they are pretty minor, which was reassuring, but when reading the main text I didn’t realize that so then I was a bit more worried (perhaps due to seeing so many papers in the past where there are serious deviations that are not justified or discussed). Perhaps this could be clarified in a footnote in the main text where the deviations are very briefly described?

p.9: Given potential power problems, why did you decide to have so many treatments? Some more discussion here would be great.

p.11, line 262: What is the size of this share?

p.13, line 319: I maybe misunderstand something, but a z-value of 2.287 should have a low p-value – is there an error here? (and maybe elsewhere? I haven’t checked the other ones.) The null hypothesis is no effect so even though the result is in the wrong direction, it is an effect that is statistically significant.

6. PLOS authors have the option to publish the peer review history of their article (what does this mean?). If published, this will include your full peer review and any attached files.

Reviewer #1: No

Reviewer #2: No

---

## [Author Response · Author response to Decision Letter 0]

14 Jul 2023

Dear Editors,

We thank you and the two anonymous reviewers for their helpful comments that substantially improved our manuscript. We have edited our submission to address all suggestions and concerns that were raised during the review process. 

Our resubmission includes a detailed point-by-point response. 

We hope the revised manuscript is now suitable for publication in PLOS One and are very much looking forward to your decision.

Thank you and best wishes also on behalf of my co-authors,

Daniel Krähmer

---

## [Editor Report · Decision Letter 1]

18 Jul 2023

Care to share? Experimental evidence on code sharing behavior in the social sciences

PONE-D-23-09667R1

Dear Dr. Krähmer,

We’re pleased to inform you that your manuscript has been judged scientifically suitable for publication and will be formally accepted for publication once it meets all outstanding technical requirements.

Kind regards,

Simon Porcher

Academic Editor

PLOS ONE
---

## [Editor Report · Acceptance letter]

28 Jul 2023

PONE-D-23-09667R1 

Care to share? Experimental evidence on code sharing behavior in the social sciences 

Dear Dr. Krähmer:

I'm pleased to inform you that your manuscript has been deemed suitable for publication in PLOS ONE. Congratulations! Your manuscript is now with our production department. 

Kind regards, 

on behalf of

Pr. Simon Porcher 

Academic Editor

PLOS ONE